# Accounting for long-range correlations in genome-wide simulations of large cohorts

**Dominic Nelson**[1], **Jerome Kelleher**[2], **Aaron P. Ragsdale**[1], **Claudia Moreau**[3], **Gil McVean**[2], **Simon Gravel**[1]*

**1** McGill University and Genome Québec Innovation Centre, McGill University, Montréal, Québec, Canada, **2** Big Data Institute, Li Ka Shing Centre for Health Information and Discovery, University of Oxford, Oxford, United Kingdom, **3** Centre Intersectoriel en Santé Durable, Université du Québec à Chicoutimi, Saguenay, Québec, Canada

* simon.gravel@mcgill.ca

**Data Availability Statement:** Software available at: https://github.com/tskit-dev/msprime.

**Funding:** This research was undertaken, in part, thanks to funding from the Canada Research

## Abstract

Coalescent simulations are widely used to examine the effects of evolution and demographic history on the genetic makeup of populations. Thanks to recent progress in algorithms and data structures, simulators such as the widely-used `msprime` now provide genome-wide simulations for millions of individuals. However, this software relies on classic coalescent theory and its assumptions that sample sizes are small and that the region being simulated is short. Here we show that coalescent simulations of long regions of the genome exhibit large biases in identity-by-descent (IBD), long-range linkage disequilibrium (LD), and ancestry patterns, particularly when the sample size is large. We present a Wright-Fisher extension to `msprime`, and show that it produces more realistic distributions of IBD, LD, and ancestry proportions, while also addressing more subtle biases of the coalescent. Further, these extensions are more computationally efficient than state-of-the-art coalescent simulations when simulating long regions, including whole-genome data. For shorter regions, efficiency can be maintained via a hybrid model which simulates the recent past under the Wright-Fisher model and uses coalescent simulations in the distant past.

## Author summary

Coalescent theory has provided deep theoretical insight into patterns of human diversity. Implementations of coalescent models in simulation software such as `ms` have further provided tools to interpret thousands of genomic studies. Recent technical progress has allowed for a dramatic increase in the scale at which genomes can be both measured and simulated, opening up opportunities for a finer understanding of evolutionary biology. However, we show that coalescent simulations of long regions of the genome exhibit large biases in sample relatedness, distorting haplotype sharing and ancestry patterns in simulated cohorts. We trace these biases to basic assumptions of the coalescent model, and show how the assumptions can be relaxed to provide a better description of the observed patterns of genetic polymorphism at a fraction of the computational cost.

Chairs program (http://www.chairs-chaires.gc.ca/)
(SG), NSERC discovery grant (http://www.nserc-crsng.gc.ca/) (SG), CIHR Discovery grant MOP-136855 (http://cihr-irsc.gc.ca/) (SG), the
Robertson Foundation (JK), the Li Ka Shing
Foundation (GM), and Wellcome Trust grant
100956/Z/13/Z (https://wellcome.ac.uk/) (GM). The
funders had no role in study design, data collection
and analysis, decision to publish, or preparation of
the manuscript.

**Competing interests:** The authors have declared
that no competing interests exist.

## Introduction

Simulations of genome evolution are widely used in the development of computational tools
for statistical and population genetics research (e.g., [1, 2, 3, 4, 5, 6]). Coalescent theory has
been used extensively for this purpose, with Hudson's ms simulation program [7] having been
cited over two thousand times since its publication in 2002. The more recent msprime coales-
cent simulation software [8] implements Hudson's original algorithm [9], but with a perfor-
mance increase of several orders of magnitude. This is achieved largely through the
introduction of a new data structure, the succinct tree sequence [10, 11], which is extremely
efficient at storing genetic variation. For example, simulating a 100 megabase region in a sam-
ple of 100,000 individuals generates an 88MB uncompressed succinct tree sequence, whereas
the Newick tree format used by ms takes approximately 3.5TB of space [8].

Simulated data are useful to the extent that they accurately reflect real genetic variation.
However, the coalescent is known to be biased relative to the Wright-Fisher model when the
sample size is large [12] or for events in the recent past [13]. However, these biases have had
limited practical impact because collecting such large empirical data sets was prohibitively
costly and the simulation of such large samples was computationally overwhelming. Both
limitations have now been lifted: sequencing datasets now regularly include thousands of
sequenced genomes, and msprime can simulate hundreds of thousands of genomes on a lap-
top computer. The assumptions of the underlying coalescent models should be carefully reex-
amined in this context.

We highlight qualitative and quantitative inaccuracies in coalescent simulations of long
regions, due to violated assumptions of the underlying genealogical model. We implement an
extension to msprime which corrects the majority of these biases via a backwards-in-time
Wright-Fisher model within msprime (see overview in Methods section and S1 Appendix),
which generates biologically plausible genealogies regardless of sample size (a separate imple-
mentation of such a model, without using succinct tree sequences, can also be found in [14]).
Our backwards-in-time Wright-Fisher simulations are also much faster than coalescent simula-
tions for large samples and long regions. For shorter regions, the coalescent is slightly faster.
Using a hybrid approach with Wright-Fisher dynamics in the recent past and coalescent dynam-
ics further back in time (as was done in [13]) preserves the computational advantages of the coa-
lescent with the long-range accuracy of the Wright-Fisher model for shorter genomic regions.

### Motivation

This work was motivated by our observation that large-scale coalescent simulations resulted in
unrealistic relatedness among samples, where nearly every pair of simulated individuals were
second- or third-degree cousins according to the time to their most recent common ancestor.
This is because individuals had too many simulated ancestors: whereas diploid individuals
carry at most $2^t$ ancestors at generation $t$ in the past, coalescent simulations allow for many
more ancestors.

This excess of ancestors is a side effect of how Hudson's coalescent algorithm models
recombination. Hudson's coalescent model assumes a small region being simulated [15], and
so does not account for multiple simultaneous recombinations during meiosis. The per-gener-
ation recombination rate in long genomic regions is maintained by multiple recombinations
occurring at different times, with each recombination introducing a new ancestral lineage.
This can lead to more than two ancestors within one generation (Fig 1).

This property of the coalescent recombination model is often innocuous when regions sim-
ulated are too short for back-and-forth recombinations to occur, or when the number of line-
ages is small enough that long range correlations are practically negligible [13, 16]. In larger

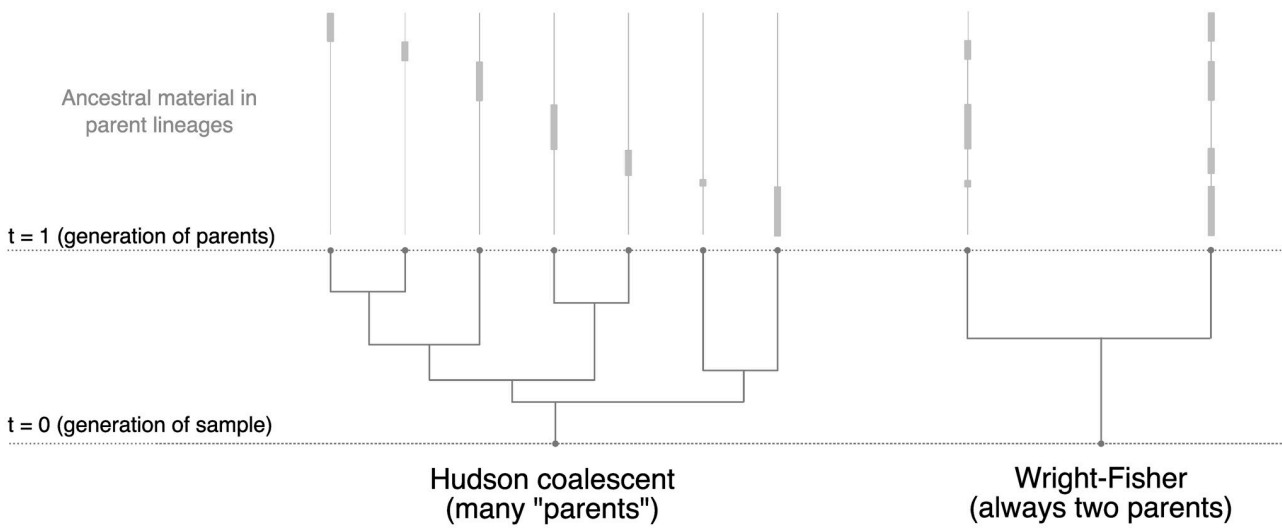

**Fig 1. Comparing coalescent and Wright-Fisher lineages one generation in the past.** A schematic of simulated lineages for a haploid sample with a single long chromosome. In the coalescent, each recombination event creates a new, independent lineage, leading to an unrealistic number of simulated parents. The Wright-Fisher model allows for back-and-forth recombination, so recombination events alternately assign genetic material between only two parental lineages. Multiple chromosomes exaggerate the difference, segregating as expected in the Wright-Fisher model but adding extra lineages under the coalescent.

samples, or under migration models, recent events induce long-range correlations along the genome [12, 17, 18, 19]. For example, individuals with a recent migrant ancestor are likely to have migrant ancestry in several chromosomes, and this is not accounted for by Hudson's coalescent. Significant differences have further been observed between the simulated genealogies of coalescent and Wright-Fisher models at a single locus [13, 14], such as the more rapid decay in the number of lineages over time in the Wright-Fisher model when sample size is large. Model differences become even more pronounced over long regions, where correlations between distant gene genealogies must be taken into account.

To highlight the magnitude of the genealogical distortions which can occur, we first use both the coalescent and Wright-Fisher models to simulate haploid sample sizes from 500 to 10,000 in a diploid population with size 10,000 and growth rate 0.001. Each sample contains 22 chromosomes of realistic lengths. Fig 2 shows that for 10,000 samples the number of lineages in the coalescent simulation increases very rapidly to reach 10 times the haploid population size $2N$ (This issue was also raised in [20, 21]). Simulations with smaller sample sizes also show a rapid growth in number of lineages to beyond the haploid population size, but the growth is slower and the excess is less pronounced than in larger samples. In the Wright-Fisher simulation, the initial growth in number of lineages is much slower and can never exceed the haploid population size, regardless of sample size.

While genealogical distortions are most clear in the first few generations, this explosion of lineages also affects genealogies in the more distant past. Fig 2 also shows that, despite rapid coalescence lowering the initial spike in the number of lineages, their number remains above the population size for hundreds of generations into the past. The effect is even more dramatic within a constant-sized population, with S2 Fig showing a case where the number of lineages remains above the effective population size for more than 100,000 generations in the past.

The number of lineages cannot be observed directly from genetic data, but these genealogical distortions have consequences for commonly used measures of genetic diversity.

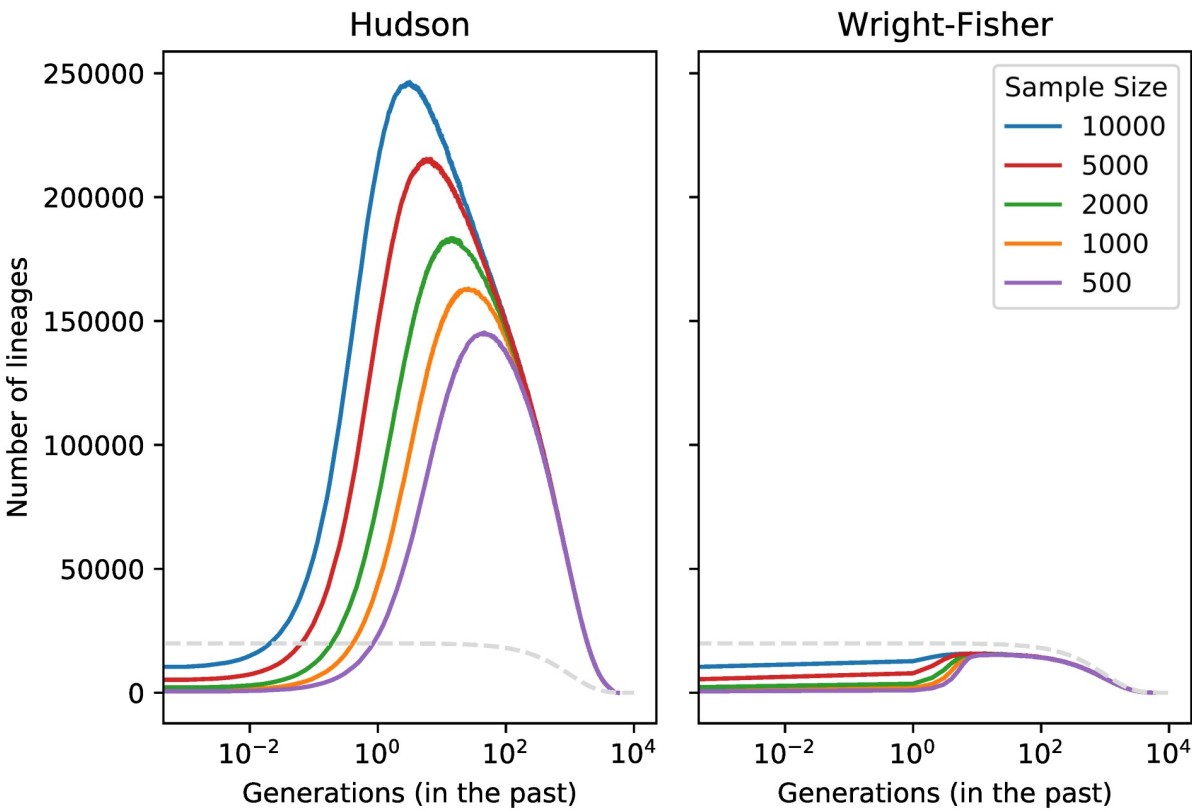

**Fig 2. Number of surviving lineages over time in coalescent and backwards-in-time Wright-Fisher dynamics.** We simulated a varying number of haploid whole genomes with 22 chromosomes of realistic lengths in a population of 10,000 diploid individuals. Dotted line shows effective population size. The implementation for simulations with multiple chromosomes is described in S1 Appendix.

## Results

In this section, we first highlight qualitative differences in multi-locus statistics between the coalescent and backwards Wright-Fisher models, and we show that the Wright-Fisher models provide a better description of the data while increasing tractability.

### Distribution of IBD

Under the Wright-Fisher model, diploid inheritance constrains the possible gene genealogies [12] and introduces correlations in IBD sharing along long simulated regions: two samples with a recent common ancestor may be IBD at several distant positions of their genome (for example on different chromosomes). In the coalescent, gene genealogies of unlinked loci are constructed independently, and do not capture this effect [12].

Modelling relatedness patterns is important in large cohorts, where cryptic relatives are common [22, 23]. To illustrate the significance of explicitly modelling diploid inheritance in a sample with close relatives, we compared simulated cohorts to genotype data from participants of the Genizon Biobank containing 8,435 individuals from the province of Quebec, Canada [24]. A description of this biobank and IBD detection methods is given in S4 Appendix. Pairwise IBD patterns observed in this cohort are shown in Fig 3.

We simulated 5,000 human haploid whole genomes (chromosome lengths and recombination rates are described in S1 Appendix) in a diploid population of constant size 10,000 under

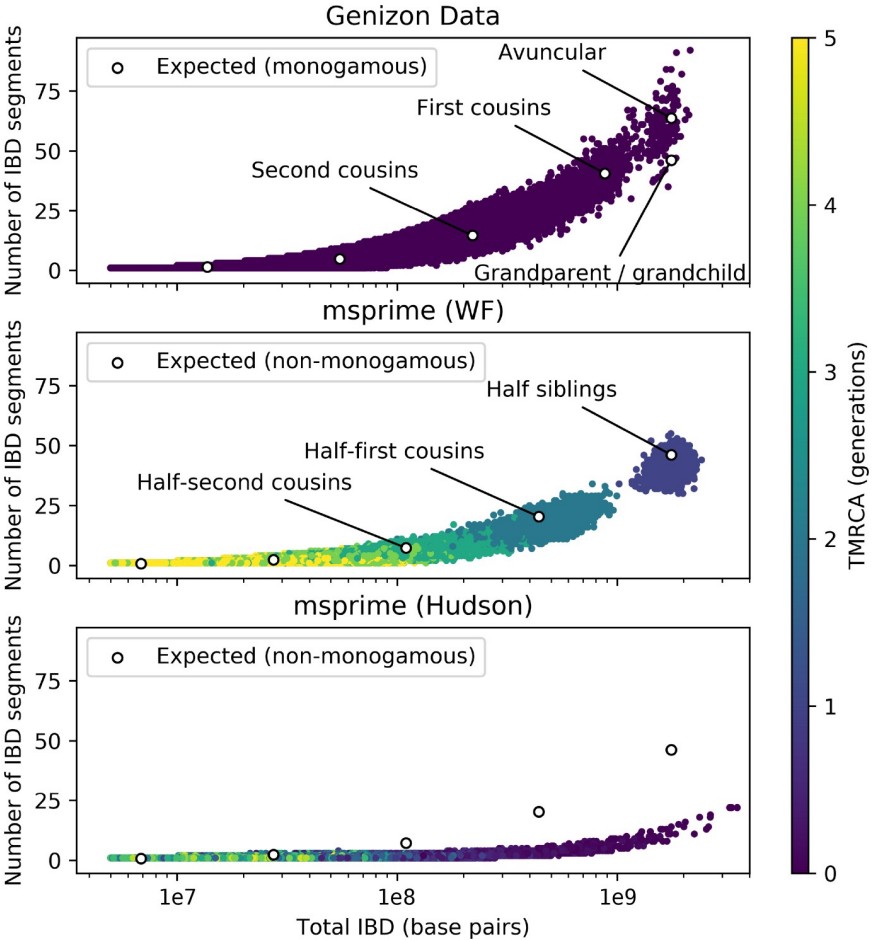

**Fig 3. Number of IBD segments between pairs of individuals versus total length of shared IBD segments.** 22 chromosomes of realistic lengths, simulated under Wright-Fisher model (middle) and coalescent (bottom), compared to data from 8,435 individuals from the Genizon Biobank (top), as well as the analytical expectation under Eqs (1), (2), (3), and (4) in S3 Appendix (white circles). Siblings were filtered from the Genizon cohort, as explained in S4 Appendix. Simulations contained 5,000 haploid samples with a diploid population size of 10,000. The isolated cluster in the Wright-Fisher simulations reflects the discrete nature of possible genealogical relationships (siblings, cousins, etc.) in the Wright-Fisher model.

the coalescent and Wright-Fisher models, and used the simulated genealogies to extract IBD segments inherited from common ancestors up to 5 generations in the past. Closer relatedness means more IBD segments and longer average length, leading to a relationship between number of segments and total length of IBD which is typically used in identifying relative status [22]. Since the detection of very short IBD segments is challenging in practice, we counted only simulated IBD segments greater than 5 centimorgans, in both simulations and the data.

Fig 3 shows the difference between the two models, with the Wright-Fisher model showing excellent qualitative agreement with the Genizon data. Quantitative differences are expected since simulations were performed in a non-monogamous randomly-mating population. By contrast, the coalescent model exhibits far too few IBD segments for closely related individuals and poor clustering by TMRCA. An analytical model for the expected number and length of shared ancestry segments (shown as white dots in Fig 3) is provided in S3 Appendix. The separated cluster predicted by the Wright-Fisher model represents simulated half-siblings: neither

full- nor half-siblings are present in the Genizon data. Other relationships also form clusters that overlap due to variance in amounts of genetic material shared IBD. Residual differences between Wright-Fisher simulations and theoretical predictions in Fig 3 have to do with the requirement that IBD segments be at least 5cM to be detected. Better agreement could be achieved by using a cutoff of 1cM in simulations (see S3 Fig).

The distribution of long IBD segments between related individuals is primarily determined by their degree of recent relatedness. For example, even though the population history and sampling process affects the *number* of sampled first cousins, the recent IBD relatedness *among* first cousins in large outbred populations is relatively independent of history and sampling: This is why the simulated and empirical distributions observed on Fig 3 are in good agreement despite differences in population sizes, and why the theoretical predictions that describe both are independent of the population demography. Because the number of close relatives changes with sampling and population size, the discrepancy between coalescent and Wright-Fisher models is more acute for large sample sizes (see S3 Fig and S4 Fig for simulations under different models). Yet S3 Fig shows clear differences between Wright-Fisher and coalescent models with $N_e = 10, 000$ and 500 samples. More generally, Shchur et. al. (2018) [23] calculated the expected number of $p$-th cousins in a sample of size $K$ taken from a population of effective size $N$. In a monogamous Wright-Fisher population, when $K/N = 0.2$, we expect approximately 55% of samples to have a first cousin, and 95% to have a second cousin within the cohort.

The long-range correlations induced by genealogical relatedness can also be measured as linkage disequilibrium between distant loci. This LD is used to estimate sizes of small populations in conservation genetics [25, 26]. Hudson's coalescent does not capture such LD patterns [17], whereas the Wright-Fisher extension to `msprime` predicts the patterns of LD expected under diploid mating (see S2 Appendix).

## Ancestry variance following admixture

In admixed populations, simulations should capture patterns of ancestry variation among present-day samples. The distribution of ancestry within recently admixed populations can be strongly dependent on pedigree structure [18], making coalescent simulations of these scenarios problematic.

We consider the variance of ancestry proportions following a single pulse of migration. Ancestry variance can be divided into genealogical variance and recombination variance [27]. In the first few generations after admixture, variance is driven by genealogical differences in the number of migrant ancestors of each individual. As time goes on, each present-day individual has more ancestors from the admixed generation, exponentially reducing this source of variance. After roughly 10 generations, variation in the amount of genetic material received from each migrant ancestor becomes a stronger source of variance [27].

We performed whole-genome simulations to evaluate how well the Wright-Fisher and coalescent models capture variance in ancestry. Fig 4 shows ancestry variance from simulations of 80 haploid samples in a diploid population of size 80, and a single event of 30% admixture at varying time in the past. These parameters were chosen to match those in [27], but here again the qualitative patterns depend weakly on the sample size and older demographic history. The approximate expected values are derived from an argument similar to the one presented for IBD sharing in S3 Appendix and outlined in [27].

The Wright-Fisher model captures both short- and long-term variance in ancestry, as expected. In the coalescent simulations the initial phase of genealogical variance is not present, leading to a 20-fold underestimate of the variance in ancestry. Lacking a diploid population

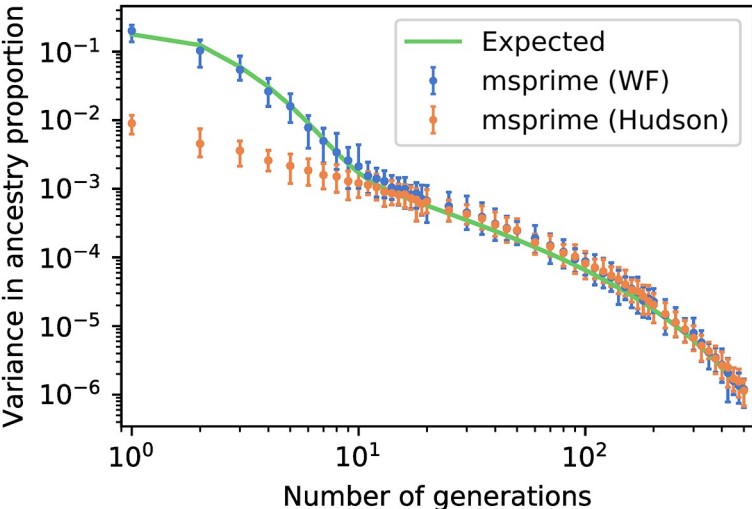

**Fig 4. Variance in ancestry after a single admixture event, as a function of time since admixture.** Calculated from 80 haploid samples in a diploid population of size 80, with 30% admixture proportions. Error bars show 95% confidence intervals over 50 simulations.

pedigree, whole-genome coalescent simulations of recently admixed populations do not reflect the distribution of ancestry expected in a large cohort, even under an idealized random-mating scenario.

## Other genealogical effects

Bhaskar et al. [13] showed that simultaneous coalescences in the Wright-Fisher model lead to more singletons and fewer doubletons than in the coalescent, which was verified in [14]. S1 Fig and S1 Table replicate these single-locus results. King et al. [17] pointed out correlation patterns among unlinked loci induced by genealogical relatedness—these results correspond to the infinite-recombination distance in S2 Appendix.

## Performance

The main advantage of `msprime` over alternate simulators is speed and scalability. This is achieved by efficient algorithms and, especially, new data structures for storing and manipulating ancestral states throughout a simulation. We therefore need to ensure that the present modification preserves these advantages.

Hudson's coalescent algorithm avoids simulating recombination and coalescent events that do not affect genetic variation in the present sample. Whereas our Wright-Fisher implementation must iterate over all discrete generations, Hudson's coalescent can traverse long stretches of time in a single step if there are no such events. The Hudson model is therefore more efficient than the Wright-Fisher model when the number of lineages is small, as can happen in small samples and short genomic regions, or in the distant past. However, Fig 2 shows that the number of lineages in whole-genome coalescent simulations is so high that the time between events is on average much less than a single generation. Furthermore, these lineages come at an additional memory and computational cost for the coalescent model. This naturally suggests using a hybrid approach with Wright-Fisher dynamics in the recent past and coalescent dynamics in the more distant past, following the approach of Bhaskar et. al. [13].

Our Wright-Fisher extension is integrated with `msprime`'s core simulation framework, and can easily be combined with coalescent simulations as part of a hybrid model. Since the optimal switching time depends on the number of extant lineages and total length of uncoalesced ancestral material, it will vary between different demographic models.

Fig 5 shows computation times for Wright-Fisher, Hudson coalescent, and hybrid simulations of 1,000 haploid samples within a population of constant size 10,000. The pure Wright-Fisher simulations are fastest at whole-genome scale, whereas pure coalescent simulations and hybrid approaches are slightly faster for shorter regions. There is a small performance cost to switching models, which explains the slightly longer runtime for the hybrid model with 100 Wright-Fisher generations versus pure coalescent simulations. The hybrid model with 1,000 Wright-Fisher generations compares favourably in terms of performance and accuracy to the coalescent for a wide range of simulated lengths.

## Methods

### Implementation

To understand the modifications needed to turn msprime into a back-in-time Wright-Fisher simulator, we first outline Hudson's original algorithm to simulate samples under the coalescent model. This brief overview is intended to give context to the modifications which enable Wright-Fisher simulations to be performed in the same framework. More details of how Hudson's algorithm is implemented in `msprime` are given in [8].

First, a number of randomly-mating populations are specified, including effective sizes and migration rates over time. Samples are introduced as haploid lineages within the populations, and the region of the genome being simulated is specified. The algorithm then constructs the

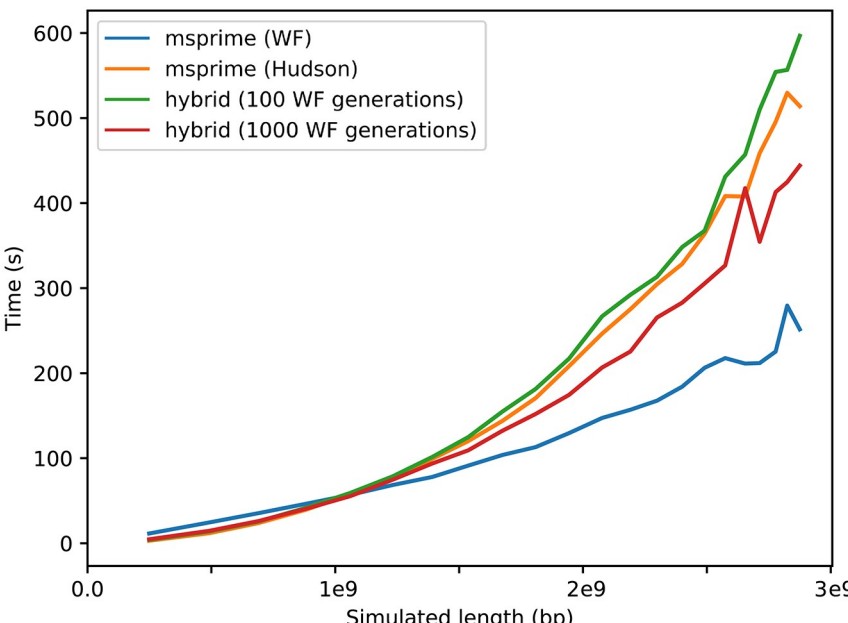

**Fig 5. Computation time of Hudson coalescent, Wright-Fisher, and hybrid models.** Hybrid models used 100 and 1000 Wright-Fisher generations before switching to the coalescent. Simulations contain from 1 to 22 chromosomes of realistic lengths (using the method described in S1 Appendix) in 1,000 haploid samples drawn from a diploid population of constant size 10,000. Results for other population sizes are shown in S5 Fig.

genealogy of each locus within this region by tracing its lineages backwards in time and tracking genomic segments that are ancestral to the sample.

To begin, each lineage contains a single ancestral segment spanning the whole simulated genomic region of a sample. As time proceeds backwards, lineages can be split by recombination events (leaving the amount of ancestral material unchanged), or participate in common ancestor events, where any overlapping regions coalesce (reducing the amount of ancestral material). The rate of recombination events depends on the sum of the genetic map distance spanned by ancestral segments carried by all extant lineages, and common ancestor events occur at a rate determined by the number of uncoalesced lineages and the effective population size. Migration events move haploid lineages between randomly-mating populations, and demographic events modify the number of populations or their size and growth rate parameters. Recombination and common ancestor events are generated at rates depending on the amount of extant ancestral material, and the simulation terminates when every position on the genome has a most recent common ancestor.

Implementing a back-in-time Wright-Fisher model requires two important changes to Hudson's algorithm. First, rather than drawing a time to the next event from an exponential distribution, we iterate though discrete generations and draw the events which occur at each time. Second, we modify the way recombination events are carried out, to account for the possibility of multiple recombinations in a single transmission: we model the number and spatial distribution of breakpoints as a Poisson process, with rate equal to the per-generation recombination rate (i.e., the distance in Morgans). This model ensures that each gamete has a unique diploid parent. An overview of this model is illustrated in Fig 1 and the detailed order of events occurring at each generation is given in S1 Appendix.

## Ethics statement

Access to the Genizon cohort genotyping data was granted under study number A07-M42-15B of the McGill university IRB. Third party data were analysed anonymously so consent was not obtained.

## Discussion

While the Wright-Fisher model may generate a more realistic pedigree than the coalescent model in the recent past, it was recognized early on as an idealized model [28, 29]. Our implementation does not track monogamous couples, for example, and therefore will vastly overestimate the prevalence of half-sibs and underestimate full sibs compared to a realistic human cohort. Assortative mating and inbreeding are not accounted for, and the migration model, while biologically plausible, is a simplification of the real migration process (see implementation details in S1 Appendix). Care should be taken in applications which are particularly sensitive to fine-scale mating or migration patterns.

Many of these issues can be addressed by allowing simulations to take place within a pre-specified pedigree, which is a natural extension to our backwards-in-time Wright-Fisher implementation. Rather than drawing genealogical links at random according to demographic parameters, lineages can simply follow a known pedigree. When reaching a pedigree founder, simulations can then continue by reverting to either the Wright-Fisher or the coalescent models. Real pedigrees of any size could then be used, from extended families up to population-scale [30], or they could be generated with the desired patterns of monogamy or assortative mating in a separate step. While conceptually straightforward, maintaining efficiency while simulating within population-scale pedigrees is non-trivial. We leave such an implementation for future work.

Improvements to recombination models is also a natural extension of the present approach. Assigning sexes to parents would allow simulation of the X-chromosome and sex-biased migration. Recombination can be extended to model crossover interference and sex-biased recombination, which have effects on the distribution of IBD [31], as well as non-crossover events.

Finally, the performance of the hybrid model could also be improved. If the number of Wright-Fisher generations were chosen optimally, it is likely to be more efficient than pure Wright-Fisher simulations in nearly all scenarios. Better guidelines for finding this optimal value could be developed, or possibly built into the simulation framework itself.

The limitations of the coalescent model have been well-studied, but were generally tied to modest effects except in very large cohorts [13]. We have shown significant qualitative and quantitative biases in whole-genome simulations of large, complex cohorts. Analysis of such cohorts is challenging, and simulations are a valuable tool for evaluating disease associations and the effects of demography in this context. We have presented here an extension to `msprime` which corrects major biases and increases performance at whole-genome scale, allowing simulations to continue supporting modern large-scale sequencing efforts.

## Supporting information

**S1 Appendix. Wright-Fisher implementation details.**
(PDF)

**S2 Appendix. Long-range linkage disequilibrium.**
(PDF)

**S3 Appendix. An approximate model for IBD sharing.**
(PDF)

**S4 Appendix. The Genizon Biobank.**
(PDF)

**S1 Table. Relative difference in mean number of singletons, doubletons, and tripletons under the Wright-Fisher ($N_{WF}$) and Hudson ($N_H$) models.**
(PDF)

**S1 Fig. Number of singletons, doubletons, and tripletons simulated under Wright-Fisher and Hudson coalescent models.** A 1Mb region was simulated 100 times in 20,000 haploid lineages in a diploid population of 10,000 individuals.
(PDF)

**S2 Fig. Number of surviving lineages over time in coalescent and back-in-time Wright-Fisher dynamics.** We simulated 10,000 haploid whole genomes with 22 chromosomes of realistic lengths in a population of 10,000 diploid individuals. The method for simulating multiple chromosomes is described in S1 Appendix. Similar results were shown in [21].
(PDF)

**S3 Fig. Number of IBD segments between pairs of individuals versus total length of shared IBD segments.** 22 chromosomes of realistic lengths, simulated under Wright-Fisher model (top) and coalescent (bottom), compared to the analytical expectation under Eqs (1) and (2) in S3 Appendix. Effective population size 10,000, sample size A) 5000, B) 2500, C) 1000, D) 500. Minimum IBD segment length of 1 centimorgan.
(PDF)

**S4 Fig. Number of IBD segments between pairs of individuals versus total length of shared IBD segments, under the Gutenkunst et. al. (2009) [3] out-of-Africa model.** 22 chromosomes of realistic lengths, simulated under Wright-Fisher model (top) and coalescent (bottom), compared to the analytical expectation under Eqs (1) and (2) in S3 Appendix. The African, European, and Asian populations had 1000 haploid samples each.
(PDF)

**S5 Fig. Computation time of Hudson coalescent, Wright-Fisher, and hybrid models with 100 and 1000 Wright-Fisher generations before switching to the coalescent.** Simulations contain from 1 to 22 chromosomes of realistic lengths, using the method described in S1 Appendix, in 500 haploid samples within a diploid population of size 500.
(PDF)

## Author Contributions

**Conceptualization:** Dominic Nelson, Jerome Kelleher, Claudia Moreau, Gil McVean, Simon Gravel.

**Data curation:** Claudia Moreau.

**Formal analysis:** Dominic Nelson, Jerome Kelleher, Aaron P. Ragsdale, Claudia Moreau, Simon Gravel.

**Funding acquisition:** Simon Gravel.

**Investigation:** Dominic Nelson, Jerome Kelleher, Gil McVean, Simon Gravel.

**Methodology:** Dominic Nelson, Jerome Kelleher, Aaron P. Ragsdale, Gil McVean, Simon Gravel.

**Software:** Dominic Nelson, Jerome Kelleher.

**Validation:** Dominic Nelson, Jerome Kelleher.

**Writing – original draft:** Dominic Nelson, Aaron P. Ragsdale, Simon Gravel.

**Writing – review & editing:** Dominic Nelson, Jerome Kelleher, Aaron P. Ragsdale, Claudia Moreau, Gil McVean, Simon Gravel.

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
