## [Decision Letter · Decision Letter 0]

4 Jul 2019

Dear Dr Nelson,

Thank you very much for submitting your Research Article entitled 'Coupling Wright-Fisher and coalescent dynamics for realistic simulation of population-scale datasets' to PLOS Genetics. Your manuscript was fully evaluated at the editorial level and by independent peer reviewers. The reviewers appreciated the attention to an important problem, but raised some substantial concerns about the current manuscript. Based on the reviews, we will not be able to accept this version of the manuscript, but we would be willing to review again a much-revised version. We cannot, of course, promise publication at that time.

Should you decide to revise the manuscript for further consideration here, your revisions should address the specific points made by each reviewer. Of note, both reviewers asked for a more detailed discussion and analysis of the impact of Wright-Fisher modeling. As noted by reviewer 2, performing simulations in a pedigree addresses the issues raised in this manuscript and should be explored. We will also require a detailed list of your responses to the review comments and a description of the changes you have made in the manuscript.

If you decide to revise the manuscript for further consideration at PLOS Genetics, please aim to resubmit within the next 60 days, unless it will take extra time to address the concerns of the reviewers, in which case we would appreciate an expected resubmission date by email to plosgenetics@plos.org.

[LINK]

We are sorry that we cannot be more positive about your manuscript at this stage. Please do not hesitate to contact us if you have any concerns or questions.

Yours sincerely,

Amy L. Williams

Guest Editor

PLOS Genetics

Hua Tang

Section Editor: Natural Variation

PLOS Genetics

Reviewer's Responses to Questions

**Comments to the Authors:**

Reviewer #1: Review of Nelson et al

In this manuscript the authors describe an implementation of a discrete time Wright Fisher model as part of the msprime package. The authors show how the coalescent approximation breaks down when the sample size, call it n, approaches the effective population size, Ne, and when large regions of the genome are simulated. While this result has been known for some time, the authors describe new facets to this issue and provide an implemented solution. The DTWF model that they implement compares favorably to the Hudson coalescent with respect to runtime and adequately captures features of the genealogical process that the coalescent approximation can not.

Generally I believe this to be a significant contribution, however the manuscript as written needs some substantial revision. I have point by point criticisms and suggestions that follow.

1) Generally it would be helpful to explore at what ratio of n/Ne these issues manifest. I would suggest revising the figures 2 and 3 to include a number of n/Ne ratios to show how this scaling effects the fit of the coalescent approximation.

2) Figure 1A is very hard to follow. It certainly does not clarify what is going on. I’d suggest the authors create a new figure to describe things.

3) Lines 87-90 in the motivation section. Is this issue purely a consequence of diploidy not being modeled? If the authors could explain the rationale a bit here it would be helpful.

4) Also with respect to motivation—the authors currently look at IBD tract lengths and the variance in ancestry as potential issues with the coalescent approximation. While this is great, both of those essentially are two facets of the same issue—recombination not being adequately captured by the coalescent. Can the authors look at different aspects of the data? For instance is the SFS perturbed in this regime under the coalescent?

5) Line 134—typo here. No section given.

6) Fig 3—unclear what the units of TMRA (shown in colors) are. Generations? Also in that figure—why is there a large gap in the data points in the top panel?

7) Figure 5 caption—the caption says that the sample was 1000 haploids but Ne=10000 diploids. Is this a typo? Was the actual sample size 10000 haploids?

8) With respect to hybrid models—it would be good to show how IBD and LD are affected by the hybrid model – are these features faithfully captured by using the hybrid models?

9) With respect to the performance analysis—it looks like the DTWF outperforms the coalescent model starting at 1e9 bp. While this is fine, we almost never have to simulate a billion bp chromosome and instead we can simulate unlinked chromosomes as separate, one from another. The authors should probably point this out.

10) In the Supplement the authors should spend more time describing the implementation. It is very non-technical at this point. Also the authors might point the reader to the code.

11) Line 382—typo “underestimated”

12) Last point—the authors should show how the issue of large samples not being adequately modeled under the coalescent is realized in empirical data. For instance the authors could analyse IBD tract lengths in the UK Biobank dataset and show that the distribution observed does not square with a coalescent process. As written the paper feels more like a technical computing note than a genetics paper.

Reviewer #2: This manuscript proposes a Wright-Fisher extension of msprime, a well-used coalescent simulator. Clearly this is a useful extension, but I feel that further work is needed for publication in PLoS Genetics.

First, it is disappointing to see only simulation results under a constant population size model. The authors should explore more realistic demographic models (e.g., previously inferred human population histories with two phases of exponential growth in the recent past) and study the accuracy of the standard coalescent model under those scenarios.

The authors have not directly demonstrated that using the WF model produces a better fit to real data. For example, it would be interesting to compare the IBD length distribution estimated from real data with simulation results from msprime (WF) and msprime (Hudson) under an inferred demographic model (e.g., inferred using the site frequency spectrum).

Co-author Kelleher has done interesting work on simulating pedigrees. It would be natural to think of a hybrid model where a pre-specified pedigree or a probabilistic pedigree model is used for the recent past, followed by the standard coalescent in the distant past. This would be a welcome extension and could be more useful than the WF extension. After all, the WF model is rather simple and idealized, while the actual mating pattern in real populations is much more complicated. Related to this point, would it be possible to incorporate other random mating models (e.g., general Cannings exchangeable models) into msprime?

Since one of the main motivations for the WF extension concerns IBD sharing, it seems important to implement crossover interference. If this is not an overly difficult extension, I would strongly recommend implementing it.

Please explain why msprime (WF) is faster than the previous version of msprime, as shown in Figure 5. Is it because the number of lineages is bounded by the population size in "msprime (WF)", as shown in Figure 2? It would be good to discuss Figure 5 in the context of Figure 2. Related to this point, please explain why "hybrid (100 WF generations)" is slower than "msprime (Hudson)", while "hybrid (1000 WF generations)" is faster than "msprime (Hudson)". To determine the optimal switch time in the hybrid model, it seems that one should investigate the trade-off between the computational overhead for using the WF model and the reduction in the number of lineages. This suggests that the optimal switch time would depend on the demographic model. This point should be clarified. Similarly, the authors should explain why "msprime (WF)" is less efficient than "msprime (Hudson)" for shorter regions, by discussing the trade-off mentioned above.

My understanding is that Bhaskar et al. (PNAS 2014, 111:2385-2390) first proposed the hybrid model, but this is not clearly acknowledged in the manuscript. The first three pages of the manuscript (including the title) give the impression that the idea is being proposed here for the first time.

Minor comments:

- Figure 1A: This figure is difficult to understand. Please explain it more clearly in the caption.

- Figure 2 : Perhaps this should be plotted with the x-axis in log scale? Also, it would not hurt to mention that the x-axis is in "Generations (backwards in time)".

- Figure 3 : In the top figure, please explain why there are few IBD segments of length between ~7*10^8 and ~10^9.

- Line 81-82: "We traced this phenomenon to samples having more than 2^t simulated ancestors at generation t in the past" is ambiguous. I think you meant, "We traced this phenomenon to some individuals in the sample having..."

- Line 95-96: It would help the reader to explain here why recent events in migration models induce long-range correlations along the genome.

- Lines 101-105: Bhaskar et al. (2014) compared the WF and the coalescent models with respect to the number of lineages at a single site. It would be good to discuss this result in relation to your result.

- There are blank references to sections throughout the manuscript. For example, Figure 2 caption ends with "described in Section ."

- Line 140: Replace "closely related samples" with "closely related individuals".

**Have all data underlying the figures and results presented in the manuscript been provided?**

Reviewer #1: Yes

Reviewer #2: None

PLOS authors have the option to publish the peer review history of their article (what does this mean?). If published, this will include your full peer review and any attached files.

Reviewer #1: No

Reviewer #2: No

---

## [Decision Letter · Decision Letter 1]

10 Dec 2019

Dear Dr Nelson,

Thank you very much for submitting your Research Article entitled 'Accounting for long-range correlations in genome-wide simulations of large cohorts' to PLOS Genetics. Your manuscript was fully evaluated at the editorial level and by independent peer reviewers. Only one comment from Reviewer 2 remains to be addressed, and this should likely be possible quickly. One possibility is to simply make a textual change.

We therefore ask you to modify the manuscript according to the review recommendations before we can consider your manuscript for acceptance. Your revisions should address the specific points made by each reviewer.

[LINK]

Yours sincerely,

Amy L. Williams

Guest Editor

PLOS Genetics

Hua Tang

Section Editor: Natural Variation

PLOS Genetics

Reviewer's Responses to Questions

**Comments to the Authors:**

Reviewer #1: I'm pleased with the edits made to this revision. This is an excellent contribution.

Reviewer #2: Overall the authors have done a good job of revising the paper and I am generally satisfied with all the changes. One exception is their response to my first major comment regarding the effect of demography on the distribution of pairwise IBD length. The authors have done simulation using the Out-of-Africa model from Gutenkunst et al. (2009), but my understanding is that in that model the present effective population sizes of YRI, CEU, and CHB are 7300, 29524, and 53403, respectively. What would happen if the present effective population size were much larger, say 1 million or 10 million, while the sample size is held at 1000? The authors claim, "the overall relationship between IBD counts and IBD length ... does not depend on the details of the demographic history or sample sizes." To me, this seems like a strong claim which warrants more rigorous justification, as it might send an incorrect message to the reader. To what extent does it not depend on the demographic model? Could you be more quantitative?

**Have all data underlying the figures and results presented in the manuscript been provided?**

Reviewer #1: Yes

Reviewer #2: None

PLOS authors have the option to publish the peer review history of their article (what does this mean?). If published, this will include your full peer review and any attached files.

Reviewer #1: No

Reviewer #2: No

---

## [Editor Report · Decision Letter 2]

21 Jan 2020

Dear Dr Nelson,

We are pleased to inform you that your manuscript entitled "Accounting for long-range correlations in genome-wide simulations of large cohorts" has been editorially accepted for publication in PLOS Genetics. Congratulations!

Yours sincerely,

Amy L. Williams

Guest Editor

PLOS Genetics

Hua Tang

Section Editor: Natural Variation

PLOS Genetics

Comments from the reviewers (if applicable):

**Data Deposition**

http://datadryad.org/submit?journalID=pgenetics&manu=PGENETICS-D-19-00848R2

**Press Queries**

---

## [Editor Report · Acceptance letter]

28 Apr 2020

PGENETICS-D-19-00848R2 

Accounting for long-range correlations in genome-wide simulations of large cohorts 

Dear Dr Nelson, 

We are pleased to inform you that your manuscript entitled "Accounting for long-range correlations in genome-wide simulations of large cohorts" has been formally accepted for publication in PLOS Genetics! Your manuscript is now with our production department and you will be notified of the publication date in due course.

With kind regards,

Matt Lyles

PLOS Genetics

On behalf of:
